# The Impact of Interval between Recurrence and Reinjection in Anti-VEGF Therapy for Diabetic Macular Edema in Pro Re Nata Regimen

**DOI:** 10.3390/jcm10245738

**Published:** 2021-12-08

**Authors:** Yoshihiro Takamura, Teruyo Kida, Hidetaka Noma, Makoto Inoue, Shigeo Yoshida, Taiji Nagaoka, Kousuke Noda, Yutaka Yamada, Masakazu Morioka, Makoto Gozawa, Takehiro Matsumura, Masaru Inatani

**Affiliations:** 1Department of Ophthalmology, Faculty of Medical Sciences, University of Fukui, Fukui 910-1193, Japan; twilightprincess0616@gmail.com (Y.Y.); mmorioka@g.u-fukui.ac.jp (M.M.); makoto.gozawa@gmail.com (M.G.); takebou_mail@yahoo.co.jp (T.M.); inatani@u-fukui.ac.jp (M.I.); 2Department of Ophthalmology, Osaka Medical and Pharmaceutical University, Takatsuki 569-8686, Japan; teruyo.kida@ompu.ac.jp; 3Department of Ophthalmology, Hachioji Medical Center, Tokyo Medical University, Hachioji 193-0998, Japan; noma-hide@umin.ac.jp; 4Department of Ophthalmology, Kyorin University School of Medicine, Tokyo 181-8611, Japan; inoue@eye-center.org; 5Department of Ophthalmology, Kurume University School of Medicine, Kurume 830-0011, Japan; yoshi@med.kurume-u.ac.jp; 6Department of Ophthalmology, Nihon University Itabashi Hospital, Tokyo 173-8610, Japan; taijinagaoka@gmail.com; 7Department of Ophthalmology, Faculty of Medicine and Graduate School of Medicine, Hokkaido University, Sapporo 060-8648, Japan; nodako@med.hokudai.ac.jp

**Keywords:** diabetic macular edema, pro re nata, VEGF, ranibizumab, aflibercept

## Abstract

Background: Pro re nata (PRN) regimen using anti-vascular endothelial growth factor (VEGF) agent is popular for the treatment of diabetic macular edema (DME). We investigated the influence of waiting time (WT) and interval between the date of recurrence of edema and re-injection on treatment efficacy. Methods: This retrospective study conducted at 7 sites in Japan enrolled patients who received intravitreal injection of ranibizumab (IVR) and aflibercept (IVA) in 1+PRN regimen. Enrolled patients were divided into 2 groups: prompt group (less than 1 week) and deferred group (3 weeks or more). Central retinal thickness (CRT) and best corrected visual acuity (BCVA) were measured every month for 1 year. Results: CRT in the deferred group was significantly higher than that in the prompt group at 2, 5, 6, 7, and 12 months (*p* < 0.05). BCVA in the prompt group was significantly better than that in the deferred group at 7, 10, and 12 months (*p* < 0.05). Conclusion: The prompt group was superior in anatomical and functional improvement of DME in anti-VEGF therapy than the deferred group. Our data suggests that shorter WT is recommended for better visual prognosis in the treatment for DME.

## 1. Introductions

Diabetic retinopathy (DR) is a main cause of visual impairment in working-age adults in Japan [1]. Diabetic macular edema (DME) commonly impairs central vision and is a clinically significant microvascular complication that can occur at any stage of DR [2]. Angiogenic mediators, such as vascular endothelial growth factor (VEGF), are recognized to play a central role in the pathogenesis of DME. Many clinical studies indicated that anti-VEGF therapy was anatomically and functionally effective, and thus, intravitreal injection of anti-VEGF agent is currently used as a gold standard of treatment for DME [3,4,5,6]. There are two types of VEGF inhibitors approved for treatment for DME in Japan, namely aflibercept and ranibizumab. Ranibizumab inhibits VEGF-A, whereas aflibercept inhibits VEGF-A, VEGF-B, and placental growth factor [7].

Macular swelling can be rapidly and dramatically improved with intravitreal injection of anti-VEGF agent; however, intraocular drug concentration decreases time dependently after a single injection, and frequent recurrences of edema are observed [8,9,10]. Thus, multiple injections of anti-VEGF agents are necessary to maintain its therapeutic effect. There is a variety of treatment regimens, such as monthly, bimonthly, pro re nata (PRN), and treat and extend (TAE) [11]. PRN involves an initial series of one or three injections, followed by further injections as deemed necessary by the ophthalmologist to treat persistent or recurrent edema. TAE is an individualized dosing scheme of titrating the injection interval based on the patient’s response of visual acuity and macular thickness. Based on a survey of Japanese retina specialists, PRN was the most common regimen in the treatment for DME [12,13].

Anti-VEGF agents have been increasingly used in clinical practice for the treatment for DME worldwide [14]. In each institution, there is a limited number of injections allowed per day; therefore, when the required treatment for patients exceeds that number, the injections are scheduled for another day. In Japan, the allowed number of injections per day varies greatly among institutions, including university hospital, private clinics, and urban and rural areas. Some facilities perform the injection immediately in response to the recurrence of DME, whereas others require the reservation of injection at a later date. We hypothesized that the longer interval between the date of the application and the actual injection for the recurrence of DME results in persistent edema, which may lead to poor visual prognosis. To verify this issue, a retrospective multicenter study was conducted to investigate the influence of the interval between the date of the application and administration of anti-VEGF drug on their efficacy to treat DME.

## 2. Materials and Methods

This retrospective study was conducted at 7 clinical centers throughout Japan. The study adhered to the tenets of the Declaration of Helsinki and was approved (IRB number: 20190127; date of approval, 19 December 2019) by the ethics committees of University of Fukui Hospital, Osaka Medical and Pharmaceutical University Hospital, Hachioji Medical Center, Kyorin University Hospital, Kurume University Hospital, Nihon University Itabashi Hospital, and Hokkaido University Hospital. This study was registered with the University Hospital Medical Information Network Clinical Trials Registry (UMIN-CTR) of Japan (ID UMIN 000039134; date of access and registration, 13 January 2020). Informed consent was obtained from the patients after the intent of the study had been fully explained. Patients with type 2 diabetes with thickening of the macular center, which was defined as central retinal thickness (CRT) of ≥300 μm in the central subfield based on spectral domain optical coherence tomography (SD-OCT) caused by DME, and who underwent intravitreal injection of ranibizumab (IVR) and aflibercept (IVA) in 1+PRN regimen, were eligible for this study. The main exclusion criteria were (1) severe DME cases with CRT of >700 μm; (2) <20 years of age; (3) focal/grid photocoagulation or panretinal photocoagulation within the previous 6 months; (4) active intraocular inflammation or infection in either eye; (5) uncontrolled glaucoma in either eye; (6) a history of intravitreal injections of steroids during the observational periods; (7) a history of stroke; (8) a systolic blood pressure (BP) of >160 mm Hg, a diastolic BP > 100 mm Hg, or untreated hypertension; and (9) glycosylated hemoglobin (HbA1c) of ≥10%. Patients who dropped out during the 12-month visit were also excluded. All patients underwent examinations, such as slit-lamp examination, dilated fundus examination, fundus photography, best corrected visual acuity (BCVA) measurement (Snellen), and intraocular pressure (IOP) measurement. BCVA measured with a Landolt chart was converted to a logarithm of the minimum angle of resolution (logMAR). After BCVA measurement, we acquired sectional and map images using SD-OCT (SPECTRALIS OCT, Heidelberg Engineering, Heidelberg, Germany) at every visit. Data on age, gender, HbA1c, and serum creatinine level were collected from medical records.

The patient was instructed to visit the hospital every month. Waiting time (WT) was defined as the period from the date of application of anti-VEGF agent injection for recurrence of DME to the date of the actual injection. According to the reservation system of the facility, the patients were divided into 2 groups: the prompt group (WT was less than 1 week) and the deferred group (WT was 3 weeks and more). Additional injection was performed if CRT exceeded 350 μm unless the patients declined it. Day 0 was defined as the day of the first injection.

Intravitreal injections were performed in a standard manner by a trained ophthalmologist using 0.4% oxybuprocaine hydrochloride (0.4% benoxyl ophthalmic solution, Santen Co. Ltd., Osaka, Japan) and 2% xylocaine as anesthetic and povidone iodine for sterilization. An eyelid speculum was used to stabilize the eyelid. The injection volume of ranibizumab (Lucentis; Novartis Pharma K.K., Tokyo, Japan) and aflibercept (Eylea; Bayer Yakuhin, Ltd. Tokyo, Japan) was 0.5 mg/0.05 mL and 2 mg/0.05 mL, respectively.

JMP (SAS Institute Inc., Tokyo, Japan) was used for statistical analyses. Data are presented as mean ± standard deviation of the mean. Wilcoxon signed-rank test was used to compare continuous variables within a group. The significance of the differences in CRT and BCVA between the groups was analyzed using Mann–Whitney *U* test. *p* < 0.05 was considered statistically significant.

## 3. Results

Of the seven institutions participating in this study, four were included in the prompt group, and three were included in the deferred group. Injection timing was delayed in the deferred group because the required number of injections exceeded the permissible number, and they were performed by appointment. We analyzed 71 eyes in the prompt group and 70 eyes in the deferred group. The average of WT was 0.55 ± 1.44 and 21.83 ± 2.07 days in the prompt and deferred groups, respectively. None of the patients experienced adverse events after injection, such as retinal detachment, endophthalmitis, neovascular glaucoma, or vitreous hemorrhage. There were no cases of cerebral and myocardial infarctions. Demographic data are presented in Table 1. There was no significant difference in age, gender, HbA1c, or serum creatinine between the two groups.

Sample cases of the prompt group and the deferred group are presented in Figure 1. In the both groups, after the injection of aflibercept, the edematous area was dramatically decreased at one month, and the recurrence was observed at two months. In the prompt group (a), when recurrence was noted at two months, an additional injection was performed on the same day. On the other hand, patients in the deferred group received the additional injection at three months (Figure 1b). Temporal profiles in the ratio of injected eyes in each month are presented in Figure 2a. The number of intravitreal injections of anti-VEGF agents in the prompt group (5.11 ± 2.59) was significantly higher than that in the deferred group (3.01 ± 1.55) (*p* < 0.0001) (Figure 2b). In the prompt group and the deferred group, 34.3% and 31.4% of eyes underwent IVR, respectively. In six eyes of the prompt group and five eyes of the deferred group, ranibizumab was switched to aflibercept during the observation periods, and in these cases, the number of injections with aflibercept was less than two. There were no cases switched from aflibercept to ranibizumab in both groups.

In both groups, CRT dramatically decreased at one month (*p* < 0.0001) after first injection, and significant reduction was noticed throughout the observational periods (Figure 3a). CRT in the deferred group was significantly higher than that in the prompt group at two months (*p* = 0.0175), five months (*p* = 0.0472), six months (*p* = 0.0172), seven months (*p* = 0.021), and 12 months (*p* = 0.014). BCVA also improved in both groups at one month and thereafter (*p* < 0.05). BCVA in the prompt group was significantly better than that in the deferred group at seven months (*p* = 0.039), 10 months (*p* = 0.046), and 12 months (*p* = 0.046) (Figure 3b). The average change of BCVA through one year were −0.14 ± 0.29 and −0.04 ± 0.36 in the prompt group and the deferred group, respectively.

## 4. Discussion

Clinicians who participated in this study performed the anti-VEGF therapy in 1+PRN regimen. According to a survey of Japanese retina specialists, 50% of respondents believed that the treat-and-extend regimen is ideal. However, the most common regimen used by 76.3% of respondents was PRN [12], and the reason was that anti-VEGF agents were expensive, and multiple injections resulted in severe financial burden on the patients. Actually, most of Japanese retina specialists consider that there is a financial problem in anti-VEGF therapy [13]. These surveys also found that most ophthalmologists prefer single injection in the loading phase and PRN for the maintenance phase. Although the importance of initial injections in the loading phase is still controversial, it is likely that the 1+PRN regimen is preferred in actual practice in Japan [13].

In this study, we investigated the influence of WT on the efficacy of anti-VEGF treatment in 1+PRN regimen. Our study found that the reduction of CRT was greater in the prompt group than the deferred group. Moreover, BCVA was also significantly better in the prompt group than that in the deferred group at several time points. These results suggest that shortening of WT contributed to better anatomical and functional improvement in the treatment for DME with anti-VEGF therapy. The longer period from the recurrence of DME to reinjection of drug implies longer exposure of retinal tissue to the status of swelling. Various factors, such as ischemia, glial reactivity, apoptosis, and photoreceptor integrity, are proposed as causes of retinal dysfunction. Persistence of edema may impair the retina and prevent the recovery of visual acuity [15,16,17].

In this study, the number of injections was higher in the prompt group than the deferred group. In the prompt group, WT was short and additional injection was promptly performed after the confirmation of recurrence. Therefore, it is inevitable that the prompt group had a higher number of injections per year than the deferred group. The more frequent number of injections of anti-VEGF agents probably contributes to better anatomical and functional recovery from DME in the prompt group. Increasing the number of injections also increases the patient’s financial burden; however, our results suggest that it is recommended to shorten the period from the recurrence of edema to reinjection to obtain a better visual outcome.

In this study, the prompt group received 5.11 ± 2.59 injections per year with change in BCVA of −0.14 ± 0.29, and the deferred group received 3.01 ± 1.55 injections with BCVA change of −0.04 ± 0.36. In the Mercury study, a recent multicenter report of PRN regimen with ranibizumab in Japan, the change in BCVA were −0.10 ± 0.24 in the eyes that underwent five or more injections and 0.03 ± 0.31 in the eyes that received 3–4 injections per year [18]. With regard to the number of injections and the change in visual acuity, the results of both studies showed comparable outcomes. Similar to the results of our study, a higher number of injections showed a better visual prognosis. In Japan, the number of injections of anti-VEGF drugs has been increasing every year, from 2.5 ± 1.8 injections in the two-year period from 2010–2012 to 5.5 ± 3.6 injections in 2015–2017 [19]. The number of injections is expected to increase further in the future, and still, the timing of injections for recurrence should not be delayed.

Differences between ranibizumab and aflibercept may influenced on the outcomes. Due to the nature of retrospective study, a few cases that were switched between the drugs were included. Thus, we could not clarify this matter, and it is the limitation of this study.

Currently, frequent administration of anti-VEGF drugs is the gold standard for DME; however, the large number of injections leads to a long interval between recurrence and re-injection, which is a problem in recent clinical practice. Based on our data, it is recommended to shorten the period from the confirmation of recurrence to reinjection. To shorten the WT, the number of injections per day and the number of days for injections should be increased. However, we must be careful not to increase the burden on ophthalmologists by increasing the number of injections, which may lead to injection-related ocular complications, such as endophthalmitis and lens damage. To improve the therapeutic efficacy of anti-VEGF agents for DME in PRN regimen, it is important to establish a system for prompt re-injections when the recurrence of edema is confirmed.

## Figures and Tables

**Figure 1 jcm-10-05738-f001:**
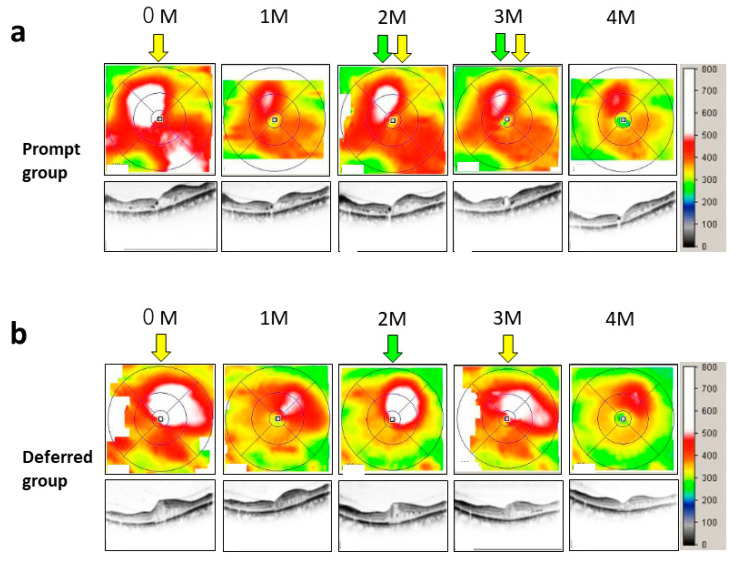
Time course of optical coherence tomography (OCT) images in thickness map and cross section was shown as the representative case in the prompt group (**a**) and in the deferred group (**b**). Green and yellow arrows indicate the timing of the application and actual injection of anti-VEGF agents, respectively, for the treatment for diabetic macular edema (DME). Scale bar indicates the retinal thickness corresponding to the false color.

**Figure 2 jcm-10-05738-f002:**
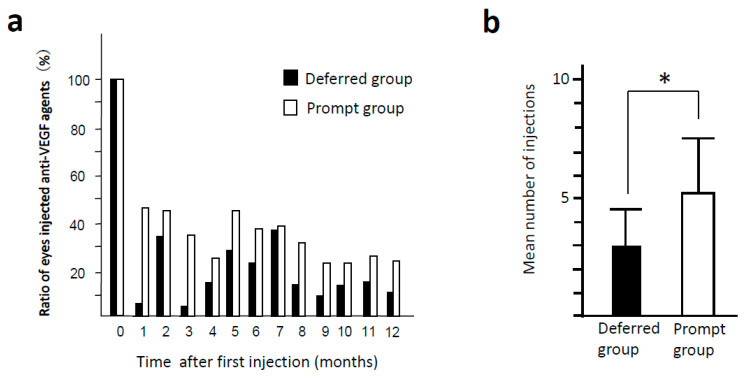
The number of injections of anti-VEGF agents in the prompt and deferred groups. (**a**) The change of the ratio of the eyes injected with anti-VEGF agents. (**b**) The mean number of injections in the prompt group was higher than that in the deferred group. * *p* < 0.05.

**Figure 3 jcm-10-05738-f003:**
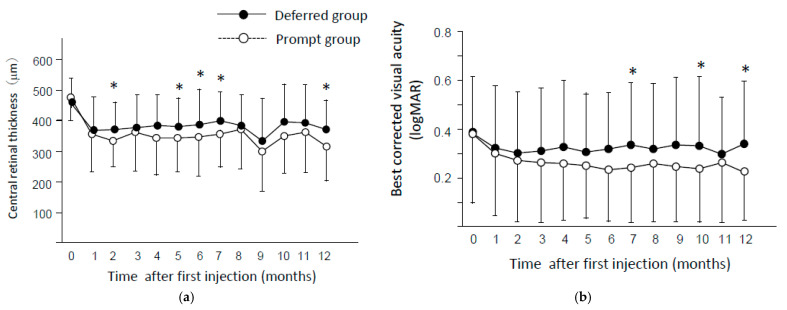
Change in (**a**) central retinal thickness (CRT) and (**b**) best corrected visual acuity (BCVA) in the prompt group and deferred group. * *p* < 0.05 (prompt group vs. deferred group). Data are presented as the mean ± standard deviations (SDs). BCVA is expressed as logMAR.

**Table 1 jcm-10-05738-t001:** Baseline characteristics at the time of registration.

	Prompt Group (*n* = 71)	Deferred Group (*n* = 70)	*p*-Value
Mean age (years)	65.1 ± 8.7	65.6 ± 9.9	0.43 ^a^
Gender (male/female)	47/24	48/22	0.54 ^b^
Left eye to right eye	37:34	32:38	0.44 ^b^
Hemoglobin A1c (%)	7.4 ± 1.5	7.5 ± 1.3	0.58 ^a^
Serum creatinine	1.07 ± 0.62	0.93 ± 0.58	0.34 ^a^

^a^ Mann–Whitney *U* test; ^b^ Chi-square test.

## Data Availability

The datasets generated during and/or analyzed during the current study are available from the corresponding author on reasonable request.

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
