# Peer review of "The Impact of Interval between Recurrence and Reinjection in Anti-VEGF Therapy for Diabetic Macular Edema in Pro Re Nata Regimen"

_jcm, 2021, doi:10.3390/jcm10245738_

Round 1
Reviewer 1 Report
The paper is an original article that compares the prompt vs delayed intravitreal anti-VEGF treatment in terms of visual acuity and central retinal thickness in patients with age related macular degenerescence treated by 1+PRN regimen. The paper is well written and the topic is of interest.
However, there are some issues that need to be addressed:
- introduction: a paragraph to present Pro re nata and treat and extend strategies would be usefull
- Line 112: Side Effects: Did you registered raised IOP and thrombotic acute events in the study group?
- Figure 3b Correction needed. In the discussion section, BCVA appears to be higher in the prompt group. However, if we consider the legend in Figure 3a it appears viceversa
- Discussion: A paragraph to compare the results of the present study with other studies should be added
Author Response
Reply for Reviewers
Dear reviewer1
We wish to express our appreciation to the Reviewer for his or her insightful comments, which have helped us significantly improve the paper. The replies for reviewer’s suggestion were as follows.
Reviewer #1:
- introduction: a paragraph to present Pro re nata and treat and extend strategies would be usefull
Based on reviewer’s suggestion, we added the following sentences in the introduction of the revised manuscript (lines 63-66).
“PRN involves an initial series of one or three injections, followed by further injections as deemed necessary by the ophthalmologist to treat persistent or recurrent edema. TAE is an individualized dosing scheme of titrating the injection interval based on the patient’s response of visual acuity and macular thickness.”
- Line 112: Side Effects: Did you registered raised IOP and thrombotic acute events in the study group?  
In this study, IOP and ischemic events were not registered as secondary outcomes. However, we confirmed that there were no cerebral and myocardial infarctions or neovascular glaucoma in this case series. We changed the sentences in the result section of the revised version, as follows. (lines 134-136)
“None of the patients experienced adverse events after injection, such as retinal detachment, endophthalmitis, neovascular glaucoma, or vitreous hemorrhage. There were no cases of cerebral and myocardial infarctions.”
- Figure 3b Correction needed. In the discussion section, BCVA appears to be higher in the prompt group. However, if we consider the legend in Figure 3a it appears viceversa
In accordance to reviewer’s suggestion, we changed from “higher” to “better” in the abstract and result section in the revised manuscript. (lines 39 and 157)
- Discussion: A paragraph to compare the results of the present study with other studies should be added
In accordance to reviewer’s suggestion, we added the following sentence in the result and discussion section in the revised version.
“The average change of BCVA through 1 year were -0.14 ± 0.29 and -0.04 ± 0.36 in the prompt group and the deferred group, respectively.” (lines 158-160)
“In this study, the prompt group received 5.11 ± 2.59 injections per year with change in BCVA of -0.14 ± 0.29, and the deferred group received 3.01 ± 1.55 injections with BCVA change of -0.04 ± 0.36. In the Mercury study, a recent multicenter report of PRN regimen with ranibizumab in Japan, the change in BCVA were -0.10 ± 0.24 in the eyes underwent 5 or more injections and 0.03 ± 0.31 in the eyes received 3-4 injections per year.18 With regard to the number of injections and the change in visual acuity, the results of both studies showed comparable outcomes. Similar to the results of our study, a higher number of injections showed a better visual prognosis. In Japan, the number of injections of anti-VEGF drugs has been increasing every year, from 2.5 ± 1.8 injections in the two-year period 2010-12 to 5.5 ± 3.6 injections in 2015-17.19 The number of injections is expected to increase further in the future, and still the timing of injections for recurrence should not be delayed.” (lines 191-201)
Also, we added 2 references as No. 18 and 19.
- Shimura M, Kitano S, Muramatsu D, et al. Real-world management of treatment-naïve diabetic macular oedema: 2-year visual outcome focusing on the starting year of intervention from STREAT-DMO study. Br J Ophthalmol. 2020;104(12):1755-1761. doi: 10.1136/bjophthalmol-2019-315726.
- Sakamoto T, Shimura M, Kitano S, et al. Impact on visual acuity and psychological outcomes of ranibizumab and subsequent treatment for diabetic macular oedema in Japan (MERCURY). Graefes Arch Clin Exp Ophthalmol. 2021. doi: 10.1007/s00417-021-05308-8.
Reviewer 2 Report
(A) Provide an overview/summary of the manuscript
To investigate the efficacy of the prompt administration of anti-VEGF for the recurrent DME during 1+PRN regimen, the retrospective study was conducted at 7 clinical centers throughout Japan. They concluded that the shorter waiting time is recommended for better visual prognosis in the treatment for DME.
(B) Introduction and discussion
The authors appropriately highlighted the aims, significance and novelty of their work. Possible influence of difference between ranibizumab and aflibercept on the results should be addressed as a limitation in Discussion.
(C) Materials and methods
The methods and statistical analyses used seem to be appropriate.
(D) Results
The reliability and validity of the results seem to be rigid.
The conclusions are supported by the data presented.
(E) Reviewer's comment
This study was well done and will give us very educational results.
Author Response
Reply for Reviewers
Dear reviewer2
We wish to express our appreciation to the Reviewer for his or her insightful comments, which have helped us significantly improve the paper. The replies for reviewer’s suggestion were as follows.
Reviewer 2
We deeply appreciate that our study was highly evaluated.
(B) Introduction and discussion
The authors appropriately highlighted the aims, significance and novelty of their work. Possible influence of difference between ranibizumab and aflibercept on the results should be addressed as a limitation in Discussion.
In accordance with reviewer’s suggestion, we added the following sentences in the result and discussion section of the revised version.
“In 6 eyes of the prompt group and 5 eyes of the deferred group, ranibizumab was switched to aflibercept during the observation periods, and in these cases the number of injections with aflibercept was less than 2. There were no cases switched from aflibercept to ranibizumab in both groups.” (lines 148-151)
“Differences between ranibizumab and aflibercept may influenced on the outcomes. Due to the nature of retrospective study, a few cases that were switched between the drugs were included. Thus, we could not clarify this matter, and it is the limitation of this study.” (lines 202-204)
Round 2
Reviewer 1 Report
The authors made the recommended changes. I have no further issues